# Effect of Cooking with Superheated (SHS) vs. Standard Steam Oven on the Fatty Acids Profile of Different Kinds of Meat and Fish

**DOI:** 10.3390/foods12040718

**Published:** 2023-02-07

**Authors:** Sara Tinagli, Roxana Elena Amarie, Giuseppe Conte, Monica Tognocchi, Marcello Mele, Alberto Mantino, Laura Casarosa, Andrea Serra

**Affiliations:** Department of Agriculture, Food and Environment University of Pisa, Via del Borghetto 80, 56124 Pisa, Italy

**Keywords:** superheated steam, fatty acids, lipid oxidation, TBARS, meat cooking

## Abstract

We compared the effect of two different kinds of steam oven—a standard (SO) and a superheated steam (SHS) oven—on four different kinds of samples: hamburgers, bovine steaks, pork steaks, and salmon fillets. Ten samples of each meat/fish were divided into three parts. Then samples were analyzed as (i) raw, (ii) cooked with SO, and (iii) cooked with SHS. For each sample, we determined the proximate composition, fatty acid composition and thiobarbituric acid reactive substances (TBARS). The results of fatty acid composition were processed both with a linear model and with a multivariate approach by using three complementary discriminant analysis techniques: canonical (CAN), stepwise (St) and discriminant (DA). SHS was effective in degreasing hamburgers but not the other kinds of samples. Cooking methods selectively affected the fatty acid profile of samples, SHS being higher in MUFA and lower in PUFA n-3 than SO. This result was also confirmed by the discriminant analysis. Finally, samples cooked with SHS showed a lower fatty acid oxidation extent than SO, as the TBARS value was significantly lower in the SHS than in the SO, irrespective of the type of meat/fish cooked.

## 1. Introduction

In the last years, meat consumption was put on trial due to its putative correlation with the risk of developing cardiovascular diseases and, for red and processed meat, colorectal cancer (CRC).

The cause of the increased risk of CRC is not yet exactly clear. Some studies [1,2] attribute it to heme iron, and others to the interaction of heme iron with added preservatives in processed meats.; cooking as well can represent a critical point that is worth monitoring. Meat is generally cooked (by boiling, steaming, grilling, or microwaving) before being eaten. Cooking makes consumption safe and improves flavor, tenderness, juiciness, and color. However, cooking causes physical and chemical changes, which can negatively affect human health. In fact, some substances are produced during cooking, such as polyaromatic hydrocarbons, heterocyclic aromatic amines, and acrylamide [3], as well as products of lipid [4] and protein [5] oxidation.

There are a lot of studies about the effect of the cooking method on the fatty acid composition of meat, meat products, and fish [4,6,7,8]. As a rule, thermal treatments determine not only the concentration of fat and fatty acids (due to water loss) but also act selectively on the fatty acid profile of the meat products. This effect depends either on the kind of cooking or fat composition. Animal species (monogastric, polygastric, and fishes), animal rearing (grazing, organic, or in a feedlot), and animal feeding (fat supplementation) [7] affect the quantity and quality of fat and especially the relative proportions of neutral and polar lipid fractions [9]. Considering the different fatty acid composition of neutral and polar lipids and that the polar fraction is usually more affected by cooking than the neutral one, thermal treatment can influence some qualitative parameters of lipids such as PUFA/SFA, the amount of PUFA n-3, the n-6/n-3 ratio, and the content of conjugated linoleic acids isomers (CLA) [4,6]. Finally, the impact of cooking on the lipids of meat can also be affected by some technological procedures; meat grinding destroys tissue integrity and dramatically increases the contact surface, and this makes the meat more prone to external physical-chemical stressors.

Cooking meat using steam preserves its chemical, nutritional, physical, and organoleptic characteristics [10]. One kind of steam cooking system is a super-heated steam oven (SHS). Currently, SHS is primarily used in the kitchens of restaurants, canteens, and hospitals. It is not yet a common domestic cooking method, but in the near future, after the optimization of the manufacturing process, a greater domestic diffusion as a high-end product is likely.

Inside SHS, there are two steam generators, which create regular steam and then superheated steam. Regular steam is created by heating water to 100 °C, and superheated steam by the regular steam passing through a second steam generator inside the oven cavity. Heat is added to the individual molecules in the second generator, giving the steam a temperature of up to 288 °C. Superheated steam is extremely unstable and seeks out the coldest area instantaneously. As a result, the steam condenses onto the food (which is the coldest area in the oven cavity), and extra heat permeates into the food. The superheated steam leads to a greater reduction in fat content compared to other types of cooking [11,12,13] while allowing the foods to maintain most of their vitamins and nutrients, which would otherwise be lost during the normal cooking process due to oxidation. In fact, SHS cooking takes place in a very low oxygen environment because the cavity of the oven is saturated by superheated steam.

Although several studies have investigated the effect of SHS cooking compared with traditional methods, such as frying, boiling, steaming, grilling, and microwaving [14]. To the best of our knowledge, there are no studies that compare SHS ovens with traditional steam ovens.

The aim of this study was to compare a traditional steam oven with an SHS oven, focusing on three main topics: (i) the degreasing effectiveness of SHS cooking, (ii) its effect on fatty acid composition with particular emphasis on fatty acids more important on nutritional and nutraceutical point of view, and (iii) its ability in preserving meat/fish from fatty acid oxidation. To provide as complete as a possible scenario with respect to the aforementioned variation factors, we studied four different kinds of samples: bovine and pork steaks, the two most popular meats from polygastric species and monogastric species, respectively, and salmon, a fish much prone to lipid oxidation and hamburgers, a product subjected to a critical procedure before cooking.

## 2. Materials and Methods

### 2.1. Experimental Design

The study was carried out on 10 samples of bovine steaks (BOV), pork steaks (POR), hamburgers (BUR), and salmon fillets (SAL). The beef and pork were obtained from Maremmana and Cinta Senese, 2 autochthonous breeds of Tuscany (Grosseto, Italy). The animals had been reared on an organic farm, “Tenuta di Paganico” (Grosseto, Italy). For the experiment, we used males receiving the same diet and slaughtered at about 750 kg (bovine) and 120 kg (pork) of live weight. From each animal, we obtained 3 steaks. The first 2 (the cooked ones) were taken from the first thoracic vertebrae of each half carcass, and the third (analyzed raw) was taken from the second thoracic vertebrae of the right half of the carcass. The steaks, about 2 cm thick, have been cooked whole (with bone and perimuscular fat).

Samples of salmon (*Salmon salar* spp.) were obtained from 10 salmon weighing about 3.5 kg live each, purchased at a local market. From each salmon, we obtained 3 portions of fillet; the cooked ones were obtained from the right and left side of the middle of each salmon, while the portion of fillet analyzed raw was sampled from the left side of each fish near where the other 2 portions had been taken.

Hamburger meat (about 300 g each) was purchased in a local market and came from different batches. Hamburger was made with a mix of 3 different bovine cuts (brisket, 40%; beef ribs, 20%; steak and flank, 35%) and 5% fat, ground and sifted with a plate with 6 mm diameter holes. Each hamburger was divided into 3 equal portions.

Thus, the experiment was carried out on 30 samples of each meat/fish divided as follows: 10 raw (R) samples, 10 cooked with a traditional steam oven (SO), and 10 samples with a superheated steam oven (SHS).

The oven used was a Demetra SHS (ATIHC S.r.l., Legnago, Verona, Italy) and a convection steam oven model Self Cooking center 61 (GN1/1) Rational (Landsberg am Lech, Germany).

The cooking programs were those indicated by the factories as the most suitable for each kind of sample analyzed, and we stopped cooking when the temperature at the center of the samples reached what is reported in Table 1. The temperature was measured by inserting a 1.3 mm diameter K-type thermocouple at the samples’ center. Data were recorded with an OM-HL-EH-TC (Omega, Stamford, CT, USA) datalogger, with an accuracy of 0.8 °C ± 0.2%, connected to a PC. Table 1 lists the size of cooked samples, the specific temperature, and the time necessary to reach the temperature at the center.

After cooking, the samples were placed in aluminum foil, packed under a vacuum to prevent oxidation, and immediately stored in a freezer (−20 °C) until analysis.

### 2.2. Analysis

The following analytical determinations were performed on each sample. Proximate composition was determined according to the AOAC procedure [15]. Dry matter analysis was performed on 5 g of the sample in a ventilated stove at 105 °C until the constant weight. Ash analysis was performed on 4 g of the sample after complete incineration of the sample and the disappearance of carbon residues in a muffle oven at 525 °C. Crude protein analysis was performed on 0.1 g of sample, according to the Kjeldahl method (N × 6.25). Ether extraction was performed on 0.5 g of the sample using an Ankom XT-10 extractor (Ankom Technology, Macedon, NY, USA).

The total lipids (TL) were extracted using a chloroform/methanol 2/1 mixture, according to Boselli et al. (2005) [16].

Fatty acid methyl esters were obtained from 10 mg of TL made to react with 0.5 N methanolic solution of sodium methoxide [17]. A total of 0.5 mg of nonadecanoic acid (C19:0) methyl ester (Sigma Chemical Co., St. Louis, MO, USA) was added as the internal standard, and the reaction was reached in 5 min at room temperature. The fatty acids were identified and quantified by injecting 1 μL of methyl esters into a GC2010 Shimadzu gas-chromatograph (Shimadzu, Columbia, MD, USA) equipped with an FID and polar capillary column (Fused Silica Capillary Column SPTM—2560, SUPELCO, Bellefonte, PA, USA, 100 m × 0.25 mm i.d., film thickness 0.20 μm). The injector temperature was set at 270 °C and the detector at 300 °C. The carrier gas was hydrogen. The flow was measured at the top of the column and was 0.7 mL/min. The injection, in split mode, took place at constant pressure. The split ratio was 1:110.

Two mixtures were used to identify the methyl esters of fatty acids. The first consisted of 37 FAME (SUPELCO, Bellefonte, PA, USA). The second mixture was used to determine the polyunsaturated fatty acids (PUFAs) consisted of non-conjugated isomers of linoleic acids, such as 5,8,11,14,17*cis*-C20:5, 4,7,10,13,16,19*cis*-C22:6 (SUPELCO, Bellefonte, PA, USA), 9,12,15*cis*-C18:3 (Matreya Inc., Pleasant Gap, PA, USA).

The identification of the C18:1 isomers was based on the use of a commercial blend (SUPELCO, Bellefonte, PA, USA) and on a comparison with the isomeric profile published by Wolff and Corrine (1995) [18]. The composition of fatty acids was expressed in grams of fatty acids in 100 g of total fatty acid (provided as Appendix A), in grams of fatty acids in 100 g of total lipids and in mg of fatty acids in 100 g of meat (provided as Appendix A).

The extent of fatty acid oxidation was measured by means of the thiobarbituric acid reactive substances (TBARS) test. Briefly, three grams of sample were homogenized in 40 mL of a 30% *w/v* solution of trichloroacetic acid (TCA) in water. After centrifugation at 5000 rpm for 15 min at 6 °C and subsequent filtration of the supernatant, 2 mL of filtered sample was reacted in a water bath at 93 °C for 45 min with 2 mL of a solution of thiobarbituric acid (TBA) at concentration 40 mM, for the development of the colorimetric reaction. After cooling, the sample was read in duplicate with the spectrophotometer at a wavelength of 532 nm. The quantification of malondialdehyde (MDA) was carried out by means of a calibration curve obtained with five standard solutions and the same number of dilutions of tetraethoxypropane and trichloroacetic acid [19]. The data are expressed in mg of MDA/kg of sample.

### 2.3. Statistical Analysis

All statistical analyses were performed with JMP Pro 16 (version 16.1.0, SAS Institute Inc., Cary, NC, USA, 1989–2021). Data were subjected to a 1-way analysis of variance as follows:y_ij_ = μ + M_i_ + C_j_ + (M_i_ × C_j_) + ε_ij_
where:y_ij_ = variable;μ = average common to all the observations;M_i_ = fixed effect ith kind of matrices with “i” variable from 1 to 4 (bovine steak, BOV; pork steak, PIG; hamburger, BUR; salmon fillet, SAL);C_j_ = fixed effect of jth kind of cooking with “j” variable from 1 to 3 (raw; cooking with superheated steam, SHS; cooking with traditional steam oven, SO);M_i_ × C_j_ = interaction of fixed factors;ε_ij_ = random error.

Data were considered statistically significant for *p* < 0.05. Data were expressed as mean + SEM. Significant interactions were tested with orthogonal contrasts between C_j_ and within M_i_.

To differentiate between the two cooking systems, we analyzed the fatty acid composition by means of the multivariate approach using 3 complementary techniques: canonical discriminant analysis (CAN), stepwise discriminant analysis (St) and discriminant analysis (DA). St allowed us to select a subset of variables that contains the minimum number of fatty acids able to discriminate two groups (SHS and SO). St was applied to 84 initial FAs, and 54 of them were retained (*p* < 0.0001). Then, the most discriminant FAs were selected and used for CAN and DA. The ability of CAN to assign each sample to the 2 groups was calculated as the percentage of correct assignments using the DA [20].

Collected data were subjected to this analysis model as follows:CAN = d_1_X_1_ + d_2_X_2_ + ... + d_n_X_n_,
where d_i_ are the canonical coefficients that indicate the contribution of each variable to the CAN, and X_i_ are the scores of the n original variables.

As the main components summarize the total variation of data, CAN summarizes variation between groups, highlighting the differences between them. In general, if n groups are involved in the study, n − 1 CAN are extracted. In our study, we considered two groups (SO and SHS); thus, 1 canonical was extracted.

The effective differentiation between groups was evaluated by using the Mahalanobis distance and the corresponding Hotelling’s T-square test [21].

## 3. Results and Discussion

### 3.1. Degreasing Effect

One of the most interesting features of SHS is the degreasing of foods and meat particularly [22]. To evaluate this effect, we performed a proximate analysis of the samples; the results are summarized in Table 2. The type of sample was a significant variation factor for all the parameters considered. In the raw samples, the total lipids differed the most between the four sample types. Hamburgers showed the highest total lipid content, followed by salmon. The total lipid of bovine and pork steaks were the same and were lower than hamburger and salmon.

The main effect of cooking was clearly a significant water loss in all four sample types. Considering the general effect of the cooking, the dry matter of 100 g of raw samples was 31.12 g, while 100 g of samples cooked with SHS and SO showed 39.13 g and 39.91 g (*p* = 0.35) of dry matter. Consequently, the other chemical components became concentrated: crude proteins in the raw samples were 19.08 g/100 g, while in SHS and SO were 22.13 and 23.13 g/100 g (*p* = 0.44); minerals in SHS and SO were 4.29 g/100 g and 4.71 g/100 g (*p* = 0.20) respectively, and 3.35 g/100 g in the raw samples.

Total lipids represented a partial exception; in fact, independently of the kind of sample, total lipids of SHS were not different from TL of raw samples (12.20 g/100 g vs. 10.80 g/100 g of sample *p* = 0.120), unlike -SO (12.80 g/100 g vs. 10.80 g/100 g of sample, *p* = 0.0130). A possible explanation for this result could be that the SHS cooking led to the loss of not only water but also fat. This result is attributable to hamburgers. Despite the binomial temperature/cooking time for the hamburgers being the lightest, the TLs cooked with SHS showed a significantly lower total lipid content than SO-BUR and raw-BUR (Table 2). Thus, the partial-degreasing ability of SHS found by other authors [11,12,13] seems confirmed, but in our study, we relieved this effect only for the hamburgers. In fact, after cooking, TLs of BOV and SAL became concentrated, while TLs of POR resulted not affected by cooking (Table 2).

This could be due to the manufacturing procedure of hamburgers which is produced with a mix of different cuts of meat and fat, including a part of intermuscular and perimuscular fat; thus, lipids are not represented only by the intramuscular fat. This makes the lipids of hamburgers more susceptible to external factors such as thermal treatment. Our results indicate that further studies are needed to assess the best conditions of binomial temperature/cooking time of the SHS to obtain the same results for all sample types.

### 3.2. Fatty Acid Composition of Raw Samples

Cooking affects not only lipid content but also lipid quality and fatty acid composition. The effect of cooking varies depending on the meat/fish being cooked and their composition in terms of fatty acids. Table 3 shows the classes of fatty acids of four raw sample types.

In terms of g/100 g of fatty acids and g/100 g of total lipids, bovine steak and hamburger had a higher content of SFA compared to pork. Salmon showed the lowest amount of SFA (Table 3). Other fatty acid classes that characterize bovine steak and hamburger are fatty acids (TFAs), both total TFA and C18:1 TFA. The most represented TFA is trans-vaccenic acid (VA, 11*t*-C18:1) which accounted for more than 53% and 60% of total TFA18:1 in BOV and BUR, respectively (Table 4). These acids are produced during rumen bio-hydrogenation of dietary unsaturated fatty acids [23]. Finally, bovine steak and hamburger contain 9*c*,11*t*-C18:2 (rumenic acid) (Table 4) and branched-chain fatty acids (BCFA) (Table 3). Rumenic acid is produced both during rumen biohydrogenation (from dietary linoleic acid) and in the tissues from desaturation of VA (synthesized in the rumen) by means of stearoyl CoA enzyme [24,25]. BCFAs are produced by using leucine, isoleucine, and valine, three amino acids coming from rumen bacteria protein degradation [26].

Salmon is a very good source of PUFAs, both omega 3 and omega 6 (Table 3). The most represented fatty acids are C18:3n-3 (alpha-linolenic acid, ALA), C20:5n-3 (EPA), C22:5n-3 and C22:6n-3 (DHA), for omega 3 and C18:2n-6, for the omega 6 (Table 4).

Finally, pork had the highest content of MUFAs (Table 3), namely 9*c*-C16:1 and 9*c*-C18:1 (oleic acid) (Table 4).

### 3.3. Effect of Cooking on Fatty Acid Composition

From a chemical point of view, SFA, 11*t*-C18:1, 9*c*,11*t*-C18:2, 9*c*-C16:1 and 9*c*-C18:1 are less prone to oxidation than EPA and DHA [27], which, in turn, have a positive nutritional impact [28,29,30].

These huge differences in fatty acid composition thus mean that our four samples can respond differently to cooking, in terms, for instance, of lipid oxidation.

Also, the “physical” structure of the meat or fish can influence the effect of cooking, which is why we compared beef steaks with hamburgers. Hamburgers are obtained from different cuts of meat and fat. A part of the fat is intramuscular fat, whereas another part is represented by intermuscular and perimuscular fat. Hamburgers are thus easier to degrease than beef steak, and this may also influence the lipid quality in terms of fatty acid composition. Several authors refer to the fact that thermal treatment selectively affects fatty acid composition, and they attribute this to a different interaction between temperature and kind of lipid [4,6,7,8]. Enser et al. 1996 [31] revealed an increase of MUFA and SFA in cooked samples, and they attribute it to the fact that neutral lipids are more prone to migration. Cobos et al. 2008 [32] report an increase in PUFA after grilling, and they related it to the drip losses of unsaturated fatty acids of triglycerides.

In our study as well, we found that the kind of cooking affected selectively fatty acid profile of samples. Table 5 lists the classes of fatty acids (expressed in g/100 g of total lipid basis) of the four kinds of samples as affected by cooking methods.

The two cooking methods affected the content of MUFA (mainly due to 9*c*-C18:1, Table 6), being higher (*p* < 0.01) in samples cooked by means of SHS (41.30 g/100 g of TL) than in that cooked by SO (38.98 g/100 g of TL), PUFA n-3, higher (*p* = 0.02) in SO (4.17 g/100 g TL) than in SHS (4.01 g/100 g TL) and, consequently, the n-6/n-3 ratio (5.21 vs. 4.42 in SHS and SO respectively, *p* = 0.04). Regarding MUFA and PUFA n-3, the interaction between the cooking method and the kind of sample was not significant (Table 5).

As a result of cooking MUFAs were higher with SHS, and PUFA n-3 were higher with SO. This seems to be unexpected, given the melting points of fatty acids. In fact, we should have found that samples cooked with SHS (more degreasing than SO) resulted in fewer PUFA n-3 and fewer MUFAs. This was true for PUFA n-3 but not for MUFA. A selective effect of cooking relating to MUFA and PUFA was also found by other authors. Jaurez et al. 2010 [6] reported a general increase in PUFA in buffalo meat cooked in three different ways, while Correia and Biscontini 2003 [33] and Rodriguez-Estrada et al. 1997 [34] refer to an increase of PUFA n-3 in grilled beef and hamburger respectively and they attributed it to triglycerides unsaturated lipid drip losses. Alfaia et al. 2010 [4], in a trial on beef cooked by means of boiling, microwaving, and grilling, found an increase of MUFA and some saturated fatty acid and a general decrease of PUFA in cooked samples, and they attributed these results also to oxidation of fatty acids. Thus, there seems to be a differential effect of the two cooking methods on the different kinds of fractions that compose total lipids. In fact, another effect of cooking was regarded the total amount of fatty acids. In samples cooked with SHS, 78.70 g/100 g of total lipids were significantly higher (*p* = 0.03) than in samples cooked with SO (75.94 g/100 g of total lipids). Given that each molecule of triglycerides contains three fatty acids while phospholipids only two, the higher the total amount of fatty acid on 100 g of total lipids, the higher the percentage of triglycerides with respect to phospholipids. This fact, together with that MUFAs are preferentially esterified into triglycerides, while PUFA n-3 are typically contained into phospholipids [35], seems to indicate that the degreasing effect of SHS was due to melting phospholipid more than proportionally than triglycerides.

Table 6 lists the fatty acids (in terms of g/100 g of total lipids) of our four kinds of samples as affected by cooking. The table highlights that the three most represented PUFA n-3, ALA, EPA and DHA, were affected by cooking but in different ways. Cooking was a significant variation factor with respect to EPA (SHS, 0.59 g/100 g of TL; SO, 0.62 g/100 g of TL; *p* = 0.04) and DHA (SHS, 1.14 g/100 g of TL; SO, 1.19 g/100 g of TL; *p* = 0.02), while the content of C18:3n3 was not different (*p* = 0.15) between samples cooked with SHS (1.52 g/100 g of TL) and SO (1.57 g/100 g of TL). EPA and DHA, which were both lower in SHS than in SO, are two typical fatty acids of phospholipids; ALA is preferentially esterified into triglycerides [36,37].

Another category of fatty acids for which cooking was a significant variation factor were TFAs, both total and C18:1 (Table 5). This is attributable mainly to VA (Table 6), the most important and represented TFA isomer (Table 3). These fatty acids are typical of bovine meat and, in pork and salmon, are present in very low amounts (Table 2 and Table 3); thus, bovine steak was the only type of sample that suffered the effect of cooking. Analogously, MUFA, TFA and VA were higher in samples cooked by means of SHS than SO (Table 5 and Table 6). TFAs and Vas, in the same way as MUFA and 9c-C18:1, are preferentially esterified into triglycerides [38]; thus, the significant effect of cooking can be explained in the same way.

To verify if the pattern of fatty acid was able to discriminate between two kinds of cooking methods, we performed a multivariate discriminant approach. Our study included two groups (SO and SHS); thus, one canonical variable was extracted. In fact, the number of CAN extracted are “n-1” where *n* is the number of groups. The selected variable had high discriminant power, with R2 > 0.3.

The canonical diagram (Figure 1) highlights that the multivariate approach was able to differentiate between the two cooking systems by using the pattern of fatty acids, SHS being separated for positive and SO for negative values. The original variables that accounted with CAN1 (scores > 1000 in absolute value) are reported in Table 7. In particular, SHS samples were characterized by C14:0, 9*c*,11*t*-C18:2, C22:3n-3, C20:5n-3 (EPA), C22:5n-3 and C22:6n-3 (DHA), while, between the fatty acids the discriminated SO it Is important to underline 9c-C14:1, 12c-C18:1, C18:3n-3, C20:2n-6 and C20:3n-3.

These results confirmed those of our univariate model about long-chain PUFA n-3 and, in particular, EPA and DHA. For the MUFAs and oleic acid, the results of the univariate and multivariate approaches were different. In the univariate approach, oleic acid and MUFAs were significantly impacted by cooking, while in the multivariate model, they were not. This is not an incongruent result as St is a multivariate technique that, step by step, reviews and re-evaluates all the variables involved and determines which one is most effective in discriminating between two groups. The contribution of an individual variable is evaluated in relation to the others to delineate the optimal variable profile to separate the groups. Consequently, it is possible that a variable may not be significant for the group’s separation in the univariate model, while it is for the multivariate approach and vice versa.

### 3.4. Effect on Fatty Acid Oxidation

Two adopted statistical approaches agreed in indicating EPA and DHA were the fatty acids most affected by different cooking methods, and as shown by the univariate approach, they were lower in SHS than in SO. Thus, it seems that the lipids of SHS samples underwent more extensive hydrolysis. Given that PUFA n-3 are very high oxidizable fatty acids [27], we should have expected SHS to be a stronger and more aggressive cooking method than SO. Conversely, our results showed that SHS was -more effective in protecting fatty acids from oxidation. In fact, TBARS values were significantly lower in SHS than in SO (0.63 vs. 0.79 ppm of Malonyldialdheyde(MDA), *p* = 0.01). Malonyldialdehyde is a very good proxy for the estimation of EPA and DHA oxidation, as it derives from the breakdown of cyclic peroxides, in turn, produced from the oxidation of fatty acids with more than two double bonds [39]. Thus, it seems that SHS melts lipids by means of the hydrolysis of triglycerides and phospholipids and the released fatty acids are lost without being oxidized, probably because SHS cooking takes place in the absence of oxygen.

Since the interaction between cooking and the sample types was not statistically significant, the SHS method appears to have a positive effect on TBARS, irrespective of whether beef, pork, hamburgers, or salmon are being cooked (Figure 2).

## 4. Conclusions

Superheated steam cooking was effective in degreasing only the hamburgers, possibly because the technological procedure for producing the hamburger made its fat more prone to melt. With respect to the fatty acid composition, samples cooked with SHS showed a lower amount of PUFA n-3 and a higher content of MUFA than samples cooked with SO, probably because SHS acted more on phospholipids than triglycerides. This determined a slight worsening of the n-6/n-3 ratio. A multivariate discriminant analysis was performed on the fatty acid composition difference between the two kinds of cooking, EPA and DHA (but not MUFA and oleic acid), and the fatty acids were most important in group separation. EPA and DHA seem to be the fatty acids most impacted by cooking. Although SHS led to higher hydrolysis of PUFA n-3 than SO, it resulted in a more protective method against fatty acid oxidation, thus a promising cooking method to give consumers high-quality meat.

## Figures and Tables

**Figure 1 foods-12-00718-f001:**
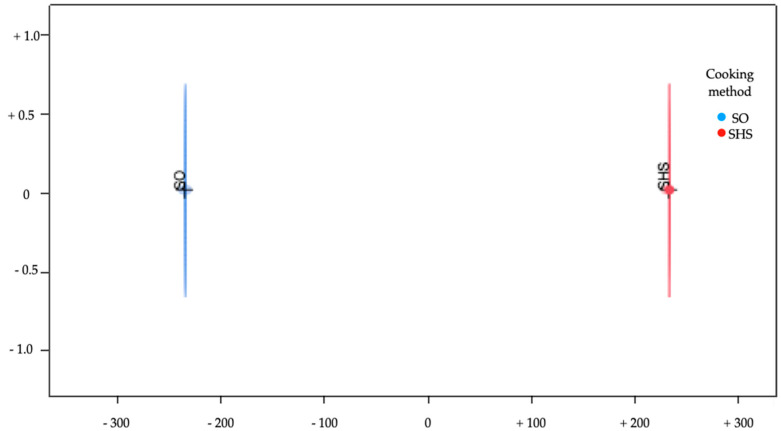
Discriminant diagram.

**Figure 2 foods-12-00718-f002:**
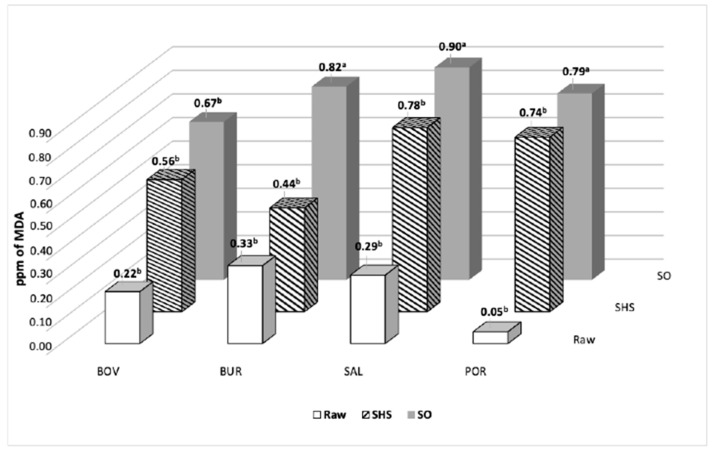
TBARS values in raw and cooked samples (ppm of MDA). BOV, bovine steak; BUR, hamburger; SAL, salmon fillet; POR, pork steak. Different letters within sample kinds correspond to different values for *p* < 0.05.

**Table 1 foods-12-00718-t001:** Experimental information.

Sample	Sample Weight (g)		Cooking Temperature °C			Temp at Center (°C)	Cooking Time (Seconds)
	SHS ^1^	SO ^2^	SHS Camber	SHS	SO Camber	SHS ^1^	SO ^2^
BOV ^3^	385.2 ± 50.5	376.6 ± 38.9	230	410	230	1231.1 ± 136.5	1265.0 ± 111.2
POR ^4^	484.4 ± 39.9	485.0 ± 35.3	230	450	220	1453.0 ± 259.8	1470.8 ± 245.9
BUR ^5^	100.3 ± 1.6	101.2 ± 0.9	240	350	240	343.6 ± 27.2	347.4 ± 29.2
SAL ^6^	666.2 ± 77.2	661.2 ± 62.2	95	320	95	792.8 ± 62.53	840 ± 144.19

^1^ SHS, SuperHeated Steam; ^2^ SO, Steam Oven; ^3^ BOV, bovine steak; ^4^ POR, pork steak; ^5^ BUR, Hamburger; ^6^ SAL, salmon fillet.

**Table 2 foods-12-00718-t002:** Proximate composition of different matrices (g/100 g of sample).

	BOV ^3^			BUR ^4^			SAL ^5^			POR ^6^			Significance		
	RAW	SHS ^1^	SO ^2^	RAW	SHS	SO	RAW	SHS	SO	RAW	SHS	SO	S ^11^	C ^12^	S × C
DM ^7^	31.67	38.77	39.15	38.78	40.80	42.17	35.80	39.16	39.87	31.32	37.79	38.46	***	***	ns
Mi ^8^	3.24	2.62	2.86	5.82	6.13	6.05	3.40	3.45	3.06	3.40	2.98	4.36	***	ns	ns
EE ^9^	6.66 ^b^	11.66 ^a^	10.69 ^a^	17.11 ^a^	14.36 ^c^	15.76 ^b^	12.94 ^b^	17.09 ^a^	17.37 ^a^	6.48	5.69	7.52	***	*	***
CP ^10^	20.39	25.20	26.49	16.90	20.71	21.27	19.52	20.43	20.05	19.53	23.37	24.70	***	***	ns

^1^ SHS, SuperHeated Steam; ^2^ SO, Steam Oven; ^3^ BOV, bovine steak; ^4^ BUR, Hamburger; ^5^ SAL, salmon fillet, ^6^ POR, pork steak; ^7^ DM, dry matter; ^8^ Mi, minerals; ^9^ EE, ether extract; ^10^ CP, crude protein, ^11^ S, sample type; ^12^ C, cooking. Different letters within sample types correspond to different values: *** *p* < 0.001; * 0.01 < *p* < 0.05; ns, not significant.

**Table 3 foods-12-00718-t003:** Classes of the fatty acid composition of raw samples (g/100 g of TL).

	BOV ^1^	HAM ^2^	SAL ^3^	POR ^4^	SE ^5^	*p*
SFA ^6^	35.85 ^a^	35.76 ^a^	12.42 ^c^	29.35 ^b^	1.40	***
MUFA ^7^	38.12 ^b^	40.47 ^b^	43.52 ^ab^	50.63 ^a^	1.46	***
PUFA ^8^	5.01 ^bc^	3.30 ^c^	28.78 ^a^	6.28 ^b^	0.58	***
n-6 ^9^	4.01 ^b^	2.01 ^c^	14.67 ^a^	5.63 ^b^	0.38	***
n-3 ^10^	0.76 ^b^	0.75 ^b^	14.01 ^a^	0.58 ^b^	0.28	***
n-6/n-3	5.25 ^b^	2.70 ^c^	1.06 ^d^	9.78 ^a^	0.17	***
TFA ^11^ _18:1_	1.03 ^b^	2.76 ^a^	0.02 ^c^	0.26 ^c^	0.07	***
TFA_total_	1.46 ^b^	3.02 ^a^	0.06 ^d^	0.31 ^c^	0.08	***
BCFA ^12^	0.61 ^b^	1.26 ^a^	0.07 ^c^	0.00 ^c^	0.06	***
Total FA	77.96	77.46	85.42	86.26	2.54	ns

^1^ BOV, bovine steak; ^2^ BUR, hamburger; ^3^ SAL, salmon; ^4^ POR, pork steak; ^5^ SE, standard error; ^6^ SFA, saturated fatty acids; ^7^ MUFA, monounsaturated fatty acids; ^8^ PUFA, polyunsaturated fatty acids; ^9^ n-6, PUFA omega 6; ^10^ n-3, PUFA omega 3; ^11^ TFA, trans fatty acids; ^12^ BCFA, branched chain fatty acid. Different letters correspond to different values: *** *p* < 0.001; ns, not significant.

**Table 4 foods-12-00718-t004:** Fatty acid composition of raw samples (g/100 g of TL).

	BOV ^1^	HAM ^2^	SAL ^3^	POR ^4^	SE ^5^	*p*
C10:0	0.07 ^a^	0.03 ^b^	0.00 ^c^	0.09 ^a^	0.00	***
C12:0	0.07 ^a^	0.06 ^ab^	0.04 ^b^	0.07 ^ab^	0.00	*
C14iso	0.04 ^b^	0.05 ^a^	0.01 ^c^	0.00 ^c^	0.00	***
C14:0	2.02 ^a^	1.98 ^a^	1.61 ^a^	1.07 ^b^	0.09	***
9*t*-C14:1	0.14 ^a^	0.01 ^b^	0.04 ^b^	0.02 ^b^	0.02	***
C15ante	0.28 ^a^	0.19 ^a^	0.00 ^b^	0.00 ^b^	0.03	***
9*c*-C14:1	0.10 ^b^	0.58 ^a^	0.00 ^b^	0.02 ^b^	0.04	***
C15:0	0.29 ^b^	0.41 ^a^	0.13 ^c^	0.03 ^d^	0.01	***
C16iso	0.16 ^b^	0.19 ^a^	0.01 ^c^	0.00 ^c^	0.01	***
C16:0	19.64 ^a^	19.56 ^a^	7.68 ^b^	19.97 ^a^	0.81	***
7*c*-C16:1	0.43 ^a^	0.00 ^c^	0.16 ^b^	0.18 ^b^	0.01	***
9*c*-C16:1	2.48 ^bc^	3.06 ^ab^	1.79 ^c^	3.70 ^a^	0.17	***
C17:0	0.73 ^a^	0.77 ^a^	0.21 ^b^	0.16 ^b^	0.02	***
C18iso	0.13 ^a^	0.03 ^bc^	0.05 ^b^	0.00 ^c^	0.01	***
9*c*-C17:1	0.45 ^a^	0.54 ^a^	0.11 ^b^	0.20 ^b^	0.02	***
C18:0	12.26 ^a^	11.54 ^a^	2.14 ^c^	7.78 ^b^	0.59	***
6/8*t*-C18:1	0.09 ^b^	0.18 ^a^	0.00 ^c^	0.06 ^b^	0.01	***
9*t*-C18:1	0.13 ^b^	0.21 ^a^	0.02 ^c^	0.11 ^b^	0.01	***
10*t*-C18:1	0.16 ^b^	0.39 ^a^	0.00 ^b^	0.09 ^b^	0.04	***
11*t*-C18:1	0.05 ^b^	1.33 ^a^	0.00 ^c^	0.00 ^c^	0.05	***
12*t*-C18:1	0.15 ^b^	0.28 ^a^	0.00 ^c^	0.00 ^c^	0.01	***
9*c*-C18:1	31.73 ^b^	30.77 ^b^	35.55 ^ab^	40.25 ^a^	1.12	***
11*c*-C18:1	1.16 ^c^	1.23 ^c^	2.79 ^b^	4.80 ^a^	0.11	***
12*c*-C18:1	0.33	0.18	0.04	0.15	0.08	ns
11*t*,15c-C18:2	0.05 ^b^	0.14 ^a^	0.00 ^c^	0.00 ^c^	0.00	***
9*c*,12*c*-C18:2	3.11 ^b^	1.68 ^c^	12.98 ^a^	4.61 ^b^	0.33	***
C20:0	0.12 ^b^	0.12 ^b^	0.25 ^a^	0.12 ^b^	0.01	***
8*c*-C20:1	0.15 ^a^	0.09 ^b^	0.11 ^b^	0.01 ^c^	0.02	***
C18:3n-3	0.45 ^b^	0.42 ^b^	4.81 ^a^	0.28 ^b^	0.08	***
9*c*,11*t*-C18:2	0.24 ^b^	0.49 ^a^	0.00 ^c^	0.04 ^c^	0.02	***
C20:2n-6	0.05 ^c^	0.02 ^c^	0.84 ^a^	0.18 ^b^	0.02	***
C20:3n-6	0.20 ^a^	0.06 ^b^	0.14 ^b^	0.07 ^a^	0.01	***
C20:3n-3	0.00 ^c^	0.01 ^c^	0.35 ^a^	0.06 ^b^	0.01	***
C20:4n-6	0.05 ^a^	0.16 ^b^	0.18 ^b^	0.59 ^a^	0.03	***
C20:5n-3	0.04 ^b^	0.02 ^b^	2.21 ^a^	0.03 ^b^	0.04	***
C22:4n-6	0.10 ^a^	0.03 ^b^	0.00 ^c^	0.12 ^a^	0.01	***
C22:5n-6	0.00 ^b^	0.01 ^b^	0.34 ^a^	0.02 ^b^	0.01	***
C22:5n-3	0.21 ^b^	0.10 ^b^	1.00 ^a^	0.12 ^b^	0.03	***
C22:6n-3	0.01 ^b^	0.01 ^b^	4.75 ^a^	0.03 ^b^	0.14	***
C20:2n-6	0.05 ^c^	0.02 ^c^	0.84 ^a^	0.18 ^b^	0.02	***
C20:3n-6	0.20 ^a^	0.06 ^b^	0.14 ^b^	0.07 ^a^	0.01	***
C20:3n-3	0.00 ^c^	0.01 ^c^	0.35 ^a^	0.06 ^b^	0.01	***
C20:4n-6	0.05 ^a^	0.16 ^b^	0.18 ^b^	0.59 ^a^	0.03	***
C20:5n-3	0.04 ^b^	0.02 ^b^	2.21 ^a^	0.03 ^b^	0.04	***
C22:4n-6	0.10 ^a^	0.03 ^b^	0.00 ^c^	0.12 ^a^	0.01	***
C22:5n-6	0.00 ^b^	0.01 ^b^	0.34 ^a^	0.02 ^b^	0.01	***
C22:5n-3	0.21 ^b^	0.10 ^b^	1.00 ^a^	0.12 ^b^	0.03	***
C22:6n-3	0.01 ^b^	0.01 ^b^	4.75 ^a^	0.03 ^b^	0.14	***

^1^ BOV, bovine steak; ^2^ BUR, hamburger; ^3^ SAL, salmon; ^4^ POR, pork steak; ^5^ SE, standard error. Different letters correspond to different values: *** *p* < 0.001; * 0.01 < *p* < 0.05; ns, not significant.

**Table 5 foods-12-00718-t005:** Classes of the fatty acid composition of different matrices (g/100 g of total lipids).

	BOV ^1^		BUR ^2^		SAL ^3^		POR ^4^		SE ^5^	Significance		
	SHS ^6^	SO ^7^	SHS ^6^	SO ^7^	SHS ^6^	SO ^7^	SHS ^6^	SO ^7^		S ^8^	C ^9^	S × C
SFA ^10^	33.79	31.17	35.97	35.16	12.13	12.20	24.78	25.52	0.81	***	ns	ns
MUFA ^11^	40.43	36.23	40.63	39.06	43.48	44.04	40.67	36.61	1.06	***	*	ns
PUFA ^12^	4.51	4.49	3.71	3.69	29.41	29.87	5.26	5.71	0.26	***	ns	ns
PUFA n6 ^13^	3.45	3.52	2.41	2.34	15.24	15.24	4.78	5.06	0.22	***	ns	ns
PUFA n3 ^14^	0.79	0.75	0.78	0.82	14.07	14.54	0.41	0.56	0.07	***	*	ns
n6/n3	4.37	4.69	3.09	2.85	1.08	1.05	11.66	9.03	0.44	***	*	ns
TFA ^15^ _18:1_	1.22 ^a^	0.97 ^b^	2.70	2.70	0.02	0.02	0.14	0.25	0.04	***	ns	*
TFA	1.68 ^a^	1.34 ^b^	2.92	2.95	0.06	0.06	0.19	0.31	0.05	***	ns	*
BCFA ^16^	0.78	0.62	1.27	1.24	0.08	0.15	0.00	0.00	0.00	***	ns	ns
Total FA	78.74	71.90	80.34	77.95	85.58	86.83	70.72	67.84	1.71	***	*	ns

^1^ BOV, bovine steak; ^2^ BUR, hamburger; ^3^ SAL, salmon; ^4^ POR, pork steak; ^5^ SE, standard error; ^6^ SHS, superheated steam oven; ^7^ SO, steam oven; ^8^ S, sample type; ^9^ C, cooking; ^10^ SFA, saturated fatty acids; ^11^ MUFA, monounsaturated fatty acids; ^12^ PUFA, polyunsaturated fatty acids, ^13^ n-6, PUFA omega 6; ^14^ n-3, PUFA omega 3; ^15^ TFA, trans fatty acids; ^16^ BCFA, branched chain fatty acid. Different letters within kinds of sample correspond to different values: *** *p* < 0.001; * 0.01 < *p* < 0.05; ns, not significant.

**Table 6 foods-12-00718-t006:** Fatty acid composition of different matrices (g/100 g of total lipids).

	BOV ^1^		BUR ^2^		SAL ^3^		POR ^4^		SE ^5^	Significance		
	SHS ^6^	SO ^7^	SHS ^6^	SO ^7^	SHS ^6^	SO ^7^	SHS ^6^	SO ^7^		S ^8^	C ^9^	S × C
C10:0	0.07	0.06	0.02	0.03	0.00	0.00	0.07	0.9	0.01	***	ns	ns
C12:0	0.08	0.06	0.06	0.06	0.04	0.03	0.05	0.07	0.00	***	ns	ns
C14iso	0.06	0.04	0.04	0.05	0.01	0.00	0.00	0.00	0.00	***	ns	ns
C14:0	1.90	1.82	1.93	1.92	1.62	1.62	0.79	0.95	0.06	***	ns	ns
9*t*-C14	0.15	0.12	0.02	0.02	0.04	0.04	0.02	0.02	0.00	***	ns	ns
C15ante	0.25	0.19	0.18	0.19	0.00	0.00	0.00	0.00	0.02	***	ns	ns
9*c*-C14-1	0.18	0.22	0.52	0.53	0.01	0.01	0.01	0.01	0.03	***	ns	ns
C15:0	0.34 ^a^	0.27 ^b^	0.40	0.40	0.13	0.12	0.02	0.03	0.01	***	ns	*
C16iso	0.19 ^a^	0.16 ^b^	0.18	0.19	0.02	0.02	0.00	0.00	0.00	***	ns	*
C16:0	18.37	17.28	19.63	19.06	7.48	7.46	16.31	16.48	0.54	***	ns	ns
7*c*-C16:1	0.49 ^a^	0.41 ^b^	0.00	0.02	0.17	0.15	0.14	0.12	0.01	***	*	**
9c-C16:1	2.63	2.36	2.88	2.83	1.81	1.82	2.88	2.52	0.13	***	ns	ns
C17:0	0.75	0.65	0.75	0.75	0.22	0.22	0.14	0.17	0.02	***	ns	ns
C18iso	0.28	0.24	0.03	0.03	0.02	0.11	0.00	0.00	0.03	***	ns	ns
9*c*-C17:1	0.31	0.28	0.50	0.52	0.14	0.09	0.17	0.16	0.04	***	ns	ns
C18:0	11.31	10.30	11.74	11.55	2.05	2.06	7.18	7.55	0.30	***	ns	ns
6/8*t*-C18:1	0.09 ^a^	0.06 ^b^	0.17	0.17	0.00	0.00	0.07 ^A^	0.04 ^B^	0.01	***	ns	*
9*t*-C18:1	0.15	0.12	0.20	0.20	0.02	0.02	0.07	0.07	0.01	***	ns	ns
10*t*-C18:1	0.12	0.11	0.37	0.38	0.00	0.00	0.00	0.00	0.01	***	ns	ns
11*t*-C18:1	0.70 ^a^	0.54 ^b^	1.32	1.31	0.00	0.00	0.00	0.02	0.03	***	ns	*
12*t*-C18:1	0.15	0.13	0.26	0.27	0.00	0.00	0.00	0.00	0.00	***	ns	ns
9*c*-C18:1	33.57	30.29	31.33	29.80	35.36	35.92	31.71	29.60	0.82	***	*	ns
11*c*-C18:1	1.44	1.26	1.27 ^a^	1.20 ^b^	2.79 ^B^	2.86 ^a^	3.69	3.03	0.09	***	*	*
12*c*-C18:1	0.14	0.12	0.18	0.17	0.05	0.04	0.12	0.10	0.01	***	*	ns
11*t*,15*c*-C18:2	0.05	0.03	0.10	0.12	0.00	0.00	0.00	0.02	0.01	***	ns	ns
9*c*,12*c*-C18:2	2.79	2.75	1.96	1.90	13.32	13.48	3.86	4.22	0.18	***	ns	ns
C20:0	0.07	0.06	0.11	0.11	0.24	0.24	0.11	0.12	0.01	***	ns	ns
8*c*-C20:1	0.15	0.13	0.08	0.08	0.12	0.12	0.00	0.00	0.01	***	ns	ns
C18:3n-3	0.51	0.44	0.43	0.43	4.98	5.11	0.18	0.31	0.04	***	ns	ns
9*c*,11*t*-C18:2	0.26	0.22	0.47	0.46	0.00	0.00	0.04	0.03	0.01	***	ns	ns
C20:2n-6	0.05	0.04	0.03	0.03	0.88	0.90	0.18	0.16	0.01	***	ns	ns
C20:3n-6	0.18	0.21	0.08	0.08	0.15	0.14	0.08	0.07	0.01	***	ns	ns
C20:3n-3	0.00	0.00	0.01	0.01	0.32 ^b^	0.39 ^a^	0.06	0.06	0.01	***	*	*
C20:4n-6	0.37	0.45	0.25	0.25	0.32	0.17	0.46	0.44	0.05	**	ns	ns
C23:0	0.02	0.02	0.02	0.02	0.05	0.04	0.00	0.00	0.00	***	ns	ns
C20:5n-3	0.02	0.04	0.04	0.04	2.28	2.37	0.02	0.03	0.02	***	ns	ns
C22:4n-6	0.06	0.06	0.05	0.04	0.02	0.00	0.11	0.08	0.01	***	ns	ns
C22:5n-6	0.00	0.00	0.01	0.00	0.33 ^b^	0.35 ^a^	0.03	0.01	0.00	***	ns	*
C22:5n-3	0.19	0.21	0.13	0.13	1.04	1.07	0.09	0.09	0.01	***	ns	ns
C22:6n-3	0.02	0.02	0.01	0.01	4.52 ^b^	4.72 ^a^	0.02	0.01	0.02	***	*	**

^1^ BOV, bovine steak; ^2^ BUR, hamburger; ^3^ SAL, salmon; ^4^ POR, pork steak; ^5^ SE, standard error; ^6^ SHS, superheated steam oven; ^7^ SO, steam oven; ^8^ S, sample type; ^9^ C, cooking. Different letters within measure unit correspond to different values: *** *p* < 0.001; ** 0.001 < *p* < 0.01; * 0.01 < *p* < 0.05; ns, not significant.

**Table 7 foods-12-00718-t007:** Scores of canonical 1.

Positive Values		Negative Values	
Fatty Acids	Scores	Fatty Acids	Scores
C21:2n-6	25,043.53	C22:4n-6	−1030.82
C22:3n-3	21,089.78	C18:3n-3	−1156.09
6/8*t*-C18:1	13,322.55	C12:0	−2292.80
6*c*-C18:1	10,301.95	13*c*-C18-1	−2323.01
C22:5n-3	9888.20	C24:0	−3090.92
C20:5n-3	8527.91	C18iso	−3094.56
C16 iso	7296.78	C17:0	−3210.54
16*t*-C18:1	6176.23	9*c*-C17:1	−3517.56
C22:5n-6	5694.56	10*t*-C18:1	−4186.55
9*c*,11*t*-C18:2	5620.77	10*t*,12*c*-C18:2	−5792.77
9*t*-C18:1	5287.86	9*c*-C14:1	−6452.71
9*t*-C16:1	5207.36	C20:2n-6	−6951.24
C22:2n-6	3832.49	C15ante	−7012.33
9*t*-C14:1	3757.06	C20:3n-3	−9031.59
12*t*-C18:1	3452.34	C10:0	−10,604.38
15*t*-C18:1	3007.15	C21:0	−10,806.21
C14:0	2681.36	12*c*-C21:1	−12,952.43
8*c*-C20:1	2361.53	12*c*-C18:1	−21,488.83
C14iso	1306.43	11*t*,13*t*-C18:2	−24,309.52
C22:6n-3	1036.20	C23:0	−28,509.83
		13*t*-C22:1	−80,814.13

## Data Availability

The data presented in this study are available on request from the corresponding author.

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
