# Peer review of "Effect of Cooking with Superheated (SHS) vs. Standard Steam Oven on the Fatty Acids Profile of Different Kinds of Meat and Fish"

_foods, 2023, doi:10.3390/foods12040718_

Round 1

Reviewer 1 Report

Comments and suggestions regarding manuscript ´Effect of cooking with superheated steam oven (SHS) on the fatty acids profile of different kind of meats and fish´ are given in attached pdf file.

Author Response

Dear Reviewer,

thanks a lot for spending your time in revision of our paper. We appreciated your suggestions and we tried to reply to all your remarks. you can find the modified text in red.

We hope the paper is now improved.

Below our punctual replies to your remarks/suggestions

L-2

R: title needs to be changed - as effect of standard steam oven was also analysed and it needs to be included in the title;

AU: title changed

L-9

R: hamburger is meat product, not kind of meat

AU: we substituted kind of meat with kind of samples

L-15

R: this does not correspond to the values presented in table 2 and statement in line 220.

AU: you are right; sorry. According also to the suggestions of reviewer 2 we rewritten this part of the abstract.

L-35

R: references regarding this need to be added;

AU: we added references of this topic.

L-43-46

R: effect of cooking on meat lipids and fatty acids should be emphasized as this is what was analysed.

AU: we modified sentence, and we had a focus on cooking and fatty acid profile (L37-50)

  1. 50

R: what about standard steam ovens? not mentioned at all... regarding price of SHS and SO ovens is it expected that they will in near future become more common as domestic cooking methode?!

AU: we provided more information about standard steam oven, and we included a sentence about the next domestic use of SHS.

L.88

R: information regarding hamburger ingredients needs to be added (percentage of meat, fat...).

why it was choosen to use hamburgers in this study, as they are meat product vs  differnt kind of raw meats?!

AU: we provided the requested information about hamburger manufacturing (L100-102). We analyzed hamburger because manufacturing procedure (grinding) can make it more prone to external stressors (such as temperature)

  1. 95.

R: detailed information regarding these programs needs to added, especially time, temperature and duration of cooking;

AU: we included this information in the table 1. The program methods were chosen to preserve the traditional organoleptic features of each sample.

L.96

R: how this temperature was determined and why exactly presented temperatures were chosen? does this means that cooking was stopped when temperature was reached regardless to duration of chosen cooking program?

AU: yes, it means. We stopped the cooking when temperature reached that reported in table; for the measure of temperature, we used a temperature probe connected with a PC. We included this information in the text (L117-119)

  1. 203 R: it is not clear what is mean by this, L. 210 R.: were the total lipids of SHS and SO significantly different? this conclusion can be missleading while presented values in the table 2 do not confirm this (except for hamburgers) and in line 220 authors stated that degreasing effect was not found. L220 R: is it the aim to ´drain´ all the fat out of steaks and to have lower fat content as imperative?!, how this would affect edible and sensory quality? And L-221 R: can same result be expected when there are different inputs (raw meat vs meat product) with different fat content and fat ´structure´ ?!

AU: we would like to reply to the remarks at lines 203, 210, 220 and 221. Please accept my apologies as I realized that this paragraph wasn’t clear enough. we Rewritten paragraph and we correct some typos in the table 2. We hope is now clearer.

In particular L-220 R: is it the aim to ´drain´ all the fat out of steaks and to have lower fat content as imperative?!, how this would affect edible and sensory quality?    

AU: it’s a partial degreasing which could affect the total calories but not the organoleptic features of samples

L-221: R:can same result be expected when there are different inputs (raw meat vs meat product) with different fat content and fat ´structure´ ?!

AU: Yes, is it. We analyzed sample from different animal species and with a different technological procedure to give as scenario as complete as possible about the different impact of cooking on fat.

L-232

R: suggestion is to present fatty acid results in only one of three ways (suggestion is to present g of fatty acids/100 g of total lipids as in table 5);

AU: thanks; suggestion accepted, and table modified.

L-277:

R: not just cooking method but also sample type and interactions SxC

AU: we modified the text.

L-285

R: it is not clear to what measure unit significance refers to... superscripts are only added for TFA18:1 and TFA

AU: in table 5 are reported the values of the interaction kind of sample x cooking method, while in the section of significance was reported the level of significance of main variation factors (samples – S- and cooking method – C). most important results relative of main variation factors are showed and disccussed in the text.

306

R: suggestion is to disscus these results with previously published papers analysing effect of different cooking methods of fat anf fatty acids.

AU: thanks for the good suggestion; we include a paragraph about the effect of cooking on fatty acids (L-323-331)

Table 6

R: there are three columns and two markings...and according to previous tables it is expected that significance was tested for sample type, cooking method and their interaction?!

AU: Sorry for the typo; we corrected the table

L-340-347

R: there are three columns and two markings...and according to previous tables it is expected that significance was tested for sample type, cooking method and their interaction?!

AU: we moved the paragraph in material and methods (L190-193)

L351

R: ordination diagram of cannonical correspondance analysis would be more inforamtive and separation of fatty acids by cooking method visible

AU: we added a table including the scores of CAN1 and commented data (L-407-411)

 L378

R: MDA is abbrevation of what?

AU: MDA means Malondialdehyde; we reported it in the text; sorry.

Figure 2

R: significantly different values shoud be marker to be visible

AU: sorry for mistake; we included letters in the figure.

L-389.

R: significantly different values shoud be marker to be visible;

AU: we modified conclusion

399-401

R: suggestion is to rather add interpretation  of nutritional quality of SHS of different meat samples and hamburger

AU: we modified conclusion

Reviewer 2 Report

Effect of cooking with superheated steam oven (SHS) on the fatty acids profile of different kind of meats and fish, it is so interest. However, there are still some problems that need to attention to improve the manuscript.

1. Title is not include the content of this paper, need improve. Such as, this paper only used two ways.

2. Please explain why used hamburgers, bovine steaks, pork steaks, and salmon fillets, and not used the chicken and so on.

3. Abstract need improve.

4. Key words: too much.

5. L 24-36 The sentences were not contact with the content of paper, need rewrite.

6. L63-71 The aim of this paper is unclear, need improve.

7. More information of Materials: hamburgers, bovine steaks, pork steaks, and salmon fillets need provide.

8. What is the number of components in this experiment

9. 3.1. Degreasing effect: The discussion need improve.

10. L340-350 The sentences need rewrite.

11. 3.4 Effect on fatty acid oxidation: The discussion need improve.

12. Tables and figures quality need improve.

13. 4. Conclusions need rewrite.

Author Response

Dear Reviewer,

thanks a lot for spending your time in revision of our paper. We appreciated your suggestions and we tried to reply to all your remarks. you can find the modified text in red.

We hope the paper is now improved.

Below our punctual replies to your remarks/suggestions

Effect of cooking with superheated steam oven (SHS) on the fatty acids profile of different kind of meats and fish, it is so interest. However, there are still some problems that need to attention to improve the manuscript.

  1. Title is not include the content of this paper, need improve. Such as, this paper only used two ways.

AU: thanks a lot. We modified the title

  1. Please explain why used hamburgers, bovine steaks, pork steaks, and salmon fillets, and not used the chicken and so on.

AU: we included and explanation as you are asking. You can find it at line 77-81 in the revised paper

  1. Abstract need improve.

AU: we modified the abstract.

  1. Key words: too much.

AU: we deleted some keywords

  1. L 24-36 The sentences were not contact with the content of paper, need rewrite.

AU: you are right. We changed the introduction according to your suggestion

  1. L63-71 The aim of this paper is unclear, need improve.

AU: we tried to make the aim clearer. You can find the modified text at lines 75-81

We modified the paragraph. We hope it is now clearer.

  1. More information of Materials: hamburgers, bovine steaks, pork steaks, and salmon fillets need provide.

AU: we provided more information

  1. What is the number of components in this experiment

Au: sorry but I can’t understand your comment.

  1. 3.1. Degreasing effect: The discussion need improve.

AU: we tried to improve the comment. We hope it is now clearer  

  1. L340-350 The sentences need rewrite.

AU:  we moved the sentence in materials and methods

  1. 3.4 Effect on fatty acid oxidation: The discussion need improve.

AU: we changed comment on fatty acids oxidation   

  1. Tables and figures quality need improve.

We corrected a typo in table 2 and table 6, We modified table 3 and 4 (according also to the suggestion of reviewer 1) we added table 7 which lists the scores of CAN1 to improve the readability of figure 1, and we add the letters for significant values in table 2.   

  1. 4. Conclusions need rewrite.

We partially changed to conclusions.

Round 2

Reviewer 1 Report

The authors have replied to all comments and suggestions with adequate changes in the manuscript, and there are no further comments and suggestions. 

Author Response

Thanks a lot for your revision and appreciating our efforts to improve the paper.

Reviewer 2 Report

I have carefully read the reviewed paper, the authors have a good answer. But also have some question in the paper.

1. 6°C or 6 °C? In all the paper.

2. Table 2. need improve. Especially, 1, 2, 3..............

3. Table 3 and 4. need improve.

Author Response

Thanks a lot for your revision. we done the changes you asked us.